

# Quantum-enhanced multiparameter estimation and compressed sensing of a field

**Youcef Baamara[1], Manuel Gessner[2,3] and Alice Sinatra[1⋆]**

**1** Laboratoire Kastler Brossel, ENS-Université PSL, CNRS, Université de la Sorbonne
and Collège de France, 24 rue Lhomond, 75231 Paris, France
**2** Departamento de Física Teórica and IFIC, Universidad de Valencia-CSIC,
C/ Dr. Moliner 50, 46100 Burjassot (Valencia), Spain
**3** ICFO-Institut de Ciències Fotòniques, The Barcelona Institute of Science and Technology,
Av. Carl Friedrich Gauss 3, 08860, Castelldefels (Barcelona), Spain

⋆ alice.sinatra@lkb.ens.fr

## Abstract

We show that a significant quantum gain corresponding to squeezed or over-squeezed spin states can be obtained in multiparameter estimation by measuring the Hadamard coefficients of a 1D or 2D signal. The physical platform we consider consists of two-level atoms in an optical lattice in a squeezed-Mott configuration, or more generally by correlated spins distributed in spatially separated modes. Our protocol requires the possibility to locally flip the spins, but relies on collective measurements. We give examples of applications to scalar or vector field mapping and compressed sensing.



## 1 Introduction

Precessing spins, or equivalently quantum systems in a superposition of two energy levels, are precise atomic sensors, and a regular spatial distribution of two-level atoms in a 1D, 2D or 3D optical lattice can be used to measure the local values of an extended field. A fundamental source of noise in such a detector is the quantum projection noise which originates from the non-commutativity of the three components of the spin 1/2 and which gives an uncertainty on the direction of the spin whose precession angle one wants to measure.

The idea of this paper is to take advantage of quantum correlations between two-level atoms in an optical lattice for multiparameter estimation, in particular for extended field measurements. To this end, besides the regular arrangement of the atoms, which offers advantages for atomic clocks [1, 2] and can be realized by means of optical tweezers or as a result of a Mott transition in a Bose-Einstein condensate [3], one should create spin correlations among the atoms. Two possible schemes, that directly yield the spin-squeezed state with one atom per site, consist in (i) adiabatically raising a lattice in a two-component Bose-Enstein condensate [4, 5] or (ii) entangling fermionic atoms located at the lattice sites via virtual tunneling processes plus an external laser which imprints a site-dependent phase [6–8]. Similar configurations but with more than one spin on each site can be obtained by splitting a spin-squeezed Bose-Einstein condensate into addressable modes [9, 10], or with atoms in a cavity where cavity-mediated interactions [11] or non-local quantum non demolition measurements [12] are used to entangle the modes. Using this last method, squeezing-enhanced distributed quantum sensing with a few modes has been recently experimentally demonstrated [12].

To take advantage of the correlations, instead of measuring the local field with one spin in each lattice site, we measure, by collective measurements involving all atoms,[1] independent linear combinations of the local fields corresponding to the Hadamard coefficients of the spatial signal discretized on the lattice. The local fields are then deduced by the inverse Hadamard transformation. For a given number of atoms and number of measurements, we then achieve a quantum gain, i.e., a reduction of the statistical uncertainty on the measured field distribution below the standard quantum limit, tracing back to the quantum correlations between the atoms. Throughout this article, the parameters are encoded in the system via local rotation generators whose directions are chosen to be the same in all sites (see (3)). In the case of a magnetic field, this is equivalent to measuring a component of the field in a chosen direction. The vector case, of a field of unknown direction and modulus, is treated in Sec. 3.2.

Spatially distributed sensors have been theoretically studied in the context of quantum multiparameter estimation, see for example [13–20] and references therein. Compared to other multiparameter quantum metrology schemes that have been proposed [17, 19], ours has the advantage that a single collective measurement has to be performed in order to obtain a given linear combination of the unknown parameters with quantum gain. Indeed, if we

---

[1]By collective measurement we mean measurement of one component of the collective spin operator ($\hat{S}_{\vec{m}}$ introduced after equation (3)), which implies that the same spin component is measured for each atom.

assume that spin flips can be performed locally [21] to select the parameter's combination, all measurements in our protocol are collective.[2] Collective measurements that are usually performed in cold atoms experiments have the advantage that they do not need the spatial resolution that would be required to perform measurements on individual atoms (i.e. local measurements).

Compared to a "scanning microscope" approach where one moves a sensor formed by an ensemble of atoms, e.g., a Bose-Einstein condensate, to probe the field locally at each site [22], our scheme, with $N$ atoms in spatially separated modes, for example in an optical lattice, to estimate $N$ parameters, gives up from the start a $1/\sqrt{N}$ factor on the standard deviation of the estimators, but it offers the advantage of using spatially separated atoms, thus without interaction. By leaving the atoms in a fixed position rather than physically scanning the trapping potential, we obtain a spatial resolution given by the wavelength of the optical lattice used to trap the atoms. Another advantage of our setup is that it naturally allows for "compressed sensing" by measuring only the first $L_{\mathcal{H}} < N$ Hadamard coefficients of the discretized field on the lattice, as we show in Sec. 4.

In the following, we develop our multiparameter estimation protocol and derive its quantum gain (Sec. 2), we study the reconstruction of a scalar or a vector field in 1D (Sec. 3), and finally, we combine our method with compressed sensing to reconstruct the field with a reduced number of measurements (Sec. 4).

## 2 Quantum enhancement in distributed sensing with collective measurements and local spin flips

We consider $N$ spins $1/2$ distributed in $N$ spatially separated modes for the estimation of $N$ parameters $\vec{\theta} = (\theta_1, ..., \theta_N)^T$ each affecting a given mode. We assume that we can manipulate the spins locally, as we could do for atoms in an optical lattice using a microscope [21], and perform collective measurements on the set of atoms. The quantum correlations between the atoms that we aim to exploit are obtained through the collective one-axis-twisting (OAT) Hamiltonian [23],

$$\hat{H}_{\text{OAT}} = \hbar \chi \left( \sum_k \hat{s}_{k,z} \right)^2 , \tag{1}$$

where $\hat{\vec{s}}_k = \hat{\vec{\sigma}}_k / 2$, $\hat{\vec{\sigma}}_k$ is the vector of the Pauli matrices for the atom in the site $k$, by evolving for a time $t$ an initial coherent spin state (CSS) with all the spins polarized along the $x$ direction

$$|\psi_0\rangle = |x\rangle^{\otimes N} . \tag{2}$$

---

[2]In multiparameter estimation theory, it is often the matter of estimating $N$ parameters by means of repeated measurements of $N$ observables of the system starting from *the same quantum state* (the observables can possibly be measured simultaneously in one realization of the experiment in case they commute). In this framework, the covariance matrix of the estimators $\Sigma$ and the quantum Fisher matrix (QFIM) $\mathcal{F}$, which are related by the multiparameter Cramér-Rao inequality [20], are usually introduced. However, as we show in section B.1, for a one-axis-twisting squeezed state it is not possible to obtain a quantum gain in the estimation of each parameter by this strategy. For a given quantum state, only one eigenvalue of $\mathcal{F}$, corresponding to a particular combination of the parameters shows a quantum advantage when all the others show a disadvantage. To obtain a quantum advantage in another combination it is necessary to change the state. For the combination that shows a quantum advantage, our scheme shows that it is sufficient to measure a single observable, which turns out to be a collective observable.

The parameters are then encoded on the state $|\psi_t\rangle = e^{-i\hat{H}_{\text{OAT}}t/\hbar}|\psi_0\rangle$ through the unitary evolution

$$\hat{U}(\vec{\theta}) = e^{-i\hat{\vec{H}}_{\vec{n}}\cdot\vec{\theta}}, \tag{3}$$

generated by the observables $\hat{\vec{H}}_{\vec{n}} = (\hat{s}_{1,\vec{n}}, ..., \hat{s}_{N,\vec{n}})^T$ with $\hat{s}_{k,\vec{n}} \equiv \vec{n}\cdot\hat{\vec{s}}_k$, where $\vec{n} = (0, n_y, n_z)^T$ is a unit vector that we consider, without loss of generality, in the plane perpendicular to the initial spin direction $x$. We consider an observable $\hat{S}_{\vec{m}}$ that is linear in the components of the collective spin $\hat{S}_{\vec{m}} = \sum_{j=1}^{N}\hat{s}_{j,\vec{m}}$ such that $\{\vec{m}, \vec{n}, \vec{e}_x\}$ form an orthonormal basis. To first order in all the $\theta_k$ near $\theta_k = 0$, its average in the state $\hat{U}(\vec{\theta})|\psi_t\rangle$ reads

$$\langle\hat{U}^{\dagger}(\vec{\theta})\hat{S}_{\vec{m}}\hat{U}(\vec{\theta})\rangle \approx -i\langle[\hat{S}_{\vec{m}}, \hat{\vec{H}}_{\vec{n}}\cdot\vec{\theta}]\rangle = -i\sum_{l,k}\theta_k\langle[\hat{s}_{l,\vec{m}}, \hat{s}_{k,\vec{n}}]\rangle\delta_{lk} = \langle\hat{s}_{1,x}\rangle\sum_k\theta_k, \tag{4}$$

where $\langle...\rangle$ denotes the average on the state $|\psi_t\rangle$ and we used the symmetry of the state. By introducing the linear combination of the parameters $\Theta \equiv \sum_k\theta_k/N$, equation (4) can be written as

$$\langle\hat{U}^{\dagger}(\vec{\theta})\hat{S}_{\vec{m}}\hat{U}(\vec{\theta})\rangle \approx \langle\hat{S}_x\rangle\Theta. \tag{5}$$

This shows that a linear observable in the collective spin components is only sensitive, to first order, to the arithmetic mean $\Theta$ of the parameters $\theta_k$. Using the single-parameter method of moments, $\Theta$ can thus be estimated by comparing the average of $\mu$ independent measurements of a linear collective spin observable $\bar{S}_{\vec{m}}^{\mu}$ with its average value $\langle\hat{S}_{\vec{m}}\rangle$ obtained theoretically or from an experimental calibration as a function of $\Theta$. In the limit $\mu \gg 1$, the method of moments allows to estimate $\Theta$ with an uncertainty $(\Delta\Theta)^2 = (\Delta\hat{S}_{\vec{m}})^2/(\mu|\partial_{\Theta}\langle\hat{S}_{\vec{m}}\rangle|^2)$ where $\partial_{\Theta} \equiv d/d\Theta$. Using the result (5), we obtain [24]

$$(\Delta\Theta)^2 = \frac{1}{\mu}\frac{(\Delta\hat{S}_{\vec{m}})^2}{|\langle\hat{S}_x\rangle|^2}. \tag{6}$$

Since the goal is to estimate all the parameters $\theta_k$ (with $k = 1, ..., N$), $N$ linearly independent combinations of the $\theta_k$ must be measured. Let us now see how, in addition to the measurement of the parameter's average $\sum_k\theta_k/N$ explained above, we can measure other linear combinations of the parameters. As we show in Appendix A, a rotation of the spin $k$ of angle $\pi$ around $x$-axis before encoding the parameter $\theta_k$ followed by a second rotation of angle $-\pi$ around the same axis after encoding the parameter, is equivalent to reversing the sign of $\theta_k$

$$e^{i\pi\hat{s}_x}e^{-i\theta\hat{s}_{\vec{n}}}e^{-i\pi\hat{s}_x} = e^{i\theta\hat{s}_{\vec{n}}}. \tag{7}$$

Let us then consider the problem of estimating $N$ parameters $\vec{\theta} = (\theta_1, ..., \theta_N)$ encoded through the unitary evolution (3), this time applying $\hat{V} = e^{-i\sum_k\alpha_k\hat{s}_{k,x}}$ and $\hat{V}^{\dagger}$ before and after the encoding of the parameters, where $\alpha_k = (1-\epsilon_k)\pi/2$ and $\epsilon_k = \pm 1$. Using (7), this can be represented by the unitary evolution

$$\hat{U}' = \hat{V}^{\dagger}e^{-i\sum_k\theta_k\hat{s}_{k,\vec{n}}}\hat{V} = \prod_k e^{i\frac{\pi}{2}(1-\epsilon_k)\hat{s}_{k,x}}e^{-i\theta_k\hat{s}_{k,\vec{n}}}e^{-i\frac{\pi}{2}(1-\epsilon_k)\hat{s}_{k,x}} = \prod_k e^{-i\epsilon_k\theta_k\hat{s}_{k,\vec{n}}} = e^{-i\sum_k\theta_k'\hat{s}_{k,\vec{n}}}, \tag{8}$$

with $\theta_k' = \epsilon_k\theta_k$, so that (8) describes the encoding of the $N$ parameters $\vec{\theta}' = (\epsilon_1\theta_1, ..., \epsilon_N\theta_N)$,

$$\hat{U}' = \hat{V}^{\dagger}\hat{U}(\vec{\theta})\hat{V} = \hat{U}(\vec{\theta}') = e^{-i\hat{\vec{H}}_{\vec{n}}\cdot\vec{\theta}'}. \tag{9}$$

Using (9) and reasoning in the same way as to obtain (5), it can be shown that to first order in the $\theta_k'$ in the vicinity of $\theta_k' = 0$, the average of $\hat{S}_{\vec{m}}$ in the state $\hat{U}'|\psi_t\rangle$ varies as

$$\langle \hat{U}'^\dagger \hat{S}_{\vec{m}} \hat{U}' \rangle = \langle \hat{S}_x \rangle \sum_k \frac{\epsilon_k \theta_k}{N} . \tag{10}$$

This last equation generalizes the result (5) and shows that, using local spin flips and the single-parameter estimation by the method of moments, the measurement of a collective spin linear observable allows to estimate the linear combination of the parameters

$$\Theta = \sum_k \epsilon_k \theta_k / N , \tag{11}$$

where $\epsilon_k = \pm 1$ with the same uncertainty (6). Note that the same calibration curve can be used for the estimation of all combinations of the parameters.

For a system in the initial CSS state, the uncertainty on the estimated combination $\Theta$ is limited by the projection noise given by $(\Delta\Theta)^2_{\mathrm{SQL}} = 1/(\mu N)$ (standard quantum limit). In the state $|\psi_t\rangle$, generated by the OAT dynamics at time $t$, it can reach a lower value $(\Delta\Theta)^2 = \xi^2/(\mu N)$ where $\xi^{-2}$ quantifies the quantum gain on the statistical error of the measurement. For a linear (L) measurement in one component $\hat{S}_{\vec{m}}$ of the collective spin, the quantum gain is limited by $\xi_{\mathrm{L}}^{-2} \leq \xi_{\mathrm{L,best}}^{-2}$ where equality is achieved for an optimal measurement direction $\vec{m} = \vec{m}_{\mathrm{L,opt}}$ and a spin squeezed state (SSS) prepared at the optimal time $t = t_{\mathrm{L,best}}$ [23,25]. One possibility to overcome the limit due to the measurement of an observable that is linear in the collective spin components, is the measurement after interaction (MAI) technique which consists in adding a second OAT evolution $\hat{U}_\tau = e^{-i\hat{H}_{\mathrm{OAT}}\tau/\hbar}$, with $\tau = -t$, after the encoding of the parameters (9) and before the measurement of the linear observable $\hat{S}_{\vec{m}}$ where $\vec{m}$ is in the $yz$-plane. This technique is equivalent to measuring a non-linear observable of the form $\hat{X}_{\mathrm{MAI}} = e^{-i\chi t \hat{S}_z^2} \hat{S}_{\vec{m}} e^{i\chi t \hat{S}_z^2}$. It turns out that this measurement is optimal in the whole time range $1/N < \chi t < 1/\sqrt{N}$ in the large $N$ limit [26,27]. Also in this case, to first order in the $\theta_k'$ in the vicinity of $\theta_k' = 0$, the average of the observable $\hat{X}_{\mathrm{MAI}}$ in the state $\hat{U}'|\psi_t\rangle$ is

$$\begin{aligned}
\langle \hat{U}'^\dagger \hat{X}_{\mathrm{MAI}} \hat{U}' \rangle &\approx \left\langle \left( \mathbb{1} + i\hat{\vec{H}}_{\vec{n}} \cdot \vec{\theta}' \right) e^{-i\chi t \hat{S}_z^2} \hat{S}_{\vec{m}} e^{i\chi t \hat{S}_z^2} \left( \mathbb{1} - i\hat{\vec{H}}_{\vec{n}} \cdot \vec{\theta}' \right) \right\rangle \\
&= -i \left\langle [e^{-i\chi t \hat{S}_z^2} \hat{S}_{\vec{m}} e^{i\chi t \hat{S}_z^2}, \hat{S}_{\vec{n}}] \right\rangle \sum_k \frac{\epsilon_k \theta_k}{N} ,
\end{aligned} \tag{12}$$

where we used the symmetry of the state $|\psi_t\rangle$. Equation (12) shows that the MAI technique allows the estimation of the linear combination $\Theta = \sum_k \epsilon_k \theta_k / N$. In an estimation protocol based on the method of moments, the uncertainty on this combination is given by [27]

$$(\Delta\Theta)^2 = \frac{1}{\mu} \frac{N/4}{|\langle [e^{i\chi t \hat{S}_z^2} \hat{S}_{\vec{m}} e^{-i\chi t \hat{S}_z^2}, \hat{S}_{\vec{n}}]\rangle|^2} . \tag{13}$$

For a time $\chi t_{\mathrm{L,best}} < \chi t \leq 1/\sqrt{N}$, the quantum gain associated with (13), with an optimal choice of $\vec{n}$ and $\vec{m}$, is larger than the gain associated with a linear measurement $\xi_{\mathrm{MAI}}^{-2} > \xi_{\mathrm{L,best}}^{-2}$ [27]. It reaches its maximum value $\xi_{\mathrm{MAI,best}}^{-2}$ at an optimal time $\chi t_{\mathrm{MAI,best}} = 1/\sqrt{N}$ in the large $N$ limit [28].

Above, we have presented the strategy that measures linear combinations of the form $\sum_k \epsilon_k \theta_k / N$, with $\epsilon_k = \pm 1$, of a set of parameters $\theta_k$ with significant quantum gain. We will now show which combinations should be measured, or which choices for $\epsilon_k$, in order to reconstruct the signal $\vec{\theta}$. A signal $\vec{\theta} = (\theta_1, ..., \theta_N)^T$ with $N = 2^m$, where $m$ is an integer, can be

decomposed in the basis of Walsh orthogonal functions: functions that take only the values $\pm 1$ represented in terms of a square matrix of order $N$ called the Hadamard matrix $\mathcal{H}_m$:

$$\theta_k = \sum_j [\mathcal{H}_m]_{kj} \tilde{\theta}_j. \tag{14}$$

The $\tilde{\theta}_j$ (for $j = 1, ..., N$) are the Hadamard coefficients associated with the signal $\vec{\theta}$, and the matrix $\mathcal{H}_m$, which satisfies the property $|[\mathcal{H}_m]_{kj}| = 1/\sqrt{N}$, is defined by recurrence with $\mathcal{H}_0 = 1$ and, for $m > 0$

$$\mathcal{H}_m = \frac{1}{\sqrt{2}} \begin{pmatrix} \mathcal{H}_{m-1} & \mathcal{H}_{m-1} \\ \mathcal{H}_{m-1} & -\mathcal{H}_{m-1} \end{pmatrix}. \tag{15}$$

The $j^{\text{th}}$ Hadamard coefficient $\tilde{\theta}_j$ is written, as a function of $\theta_k$, as

$$\tilde{\theta}_j = \sum_k [\mathcal{H}_m^{-1}]_{jk} \theta_k. \tag{16}$$

Comparing this last equation with (11) we see that for a suitable choice of $\epsilon_k = \pm 1$ one obtains

$$\sqrt{N}\Theta = \tilde{\theta}, \tag{17}$$

such that the combinations measured by our strategy are, up to a factor $\sqrt{N}$, the Hadamard coefficients of the signal $\vec{\theta}$. Once these coefficients are measured independently and with the same uncertainty, we can deduce the original signal using (14). All the measured parameters $\theta_k$ thus have the same uncertainty

$$(\Delta \theta_k)^2 = (\Delta \tilde{\theta}_k)^2 = N(\Delta\Theta)^2 = \frac{\xi^2}{\mu} \quad \forall \, k. \tag{18}$$

Unlike the estimation of a single parameter with the $N$-atom coherent spin state, the uncertainty (18) on the parameters estimated by our strategy with the CSS state is independent of the size $N$ of the system. This can be explained by the fact that each parameter $\theta_k$ is locally encoded on an individual atom. The quantum correlations between atoms generated by the OAT dynamics allow us to introduce a dependence in the system size $N$ of the uncertainties $(\Delta \theta_k)^2$ through the parameter $\xi$. As we will show in Appendix B, this strategy can also be understood in the framework of multiparameter estimation theory. The quantum metrological gain associated with each parameter is then given, as in the single parameter case, by $\xi^{-2}$. The dependence of this parameter on $N$ and $\chi t$ in the absence and presence of decoherence, with a linear measurement or by the MAI technique, is studied in detail in Ref. [27]. The question of the optimality of our strategy and a comparison with the quantum Fisher matrix in the framework of the theory of multiparameter estimation are presented in the Appendix B. In the following sections, we give two examples of the application of the method, to the mapping of a scalar and vectorial one-dimensional field, and to compressed sensing.

# 3 Mapping of a one-dimensional field: simulation with $N = 8$

## 3.1 Scalar field

We give here an illustration of the application of our strategy to the measurement of a scalar field $\theta(x)$ that varies along one direction of space. In our numerical simulation below, we consider a field of the form

$$\theta(x) = \theta_0 \sin(x), \tag{19}$$

that we discretize on $N = 8$ sites $\vec{\theta} = (\theta_1, ..., \theta_N)$ with $\theta_i \equiv \theta(x_i)$, each site having as sensor a two-level atom. We assume that the encoding of the $\vec{\theta}$ parameters is done with the $\hat{\vec{H}}_{\vec{n}} = (\hat{s}_{1,\vec{n}}, ..., \hat{s}_{N,\vec{n}})^T$ generators, $\vec{n}$ being the optimal direction in the $yz$-plane, through the unitary evolution (3). According to our protocol, to estimate the Hadamard coefficient $\tilde{\theta}_j$ after an evolution for a time $t$ with the OAT Hamiltonian (1) of the initial CSS state (2), we apply the unitary evolution

$$\hat{U}' = e^{i \sum_k \alpha_k \hat{s}_{k,x}} e^{-i \sum_k \theta_k \hat{s}_{k,\vec{n}}} e^{-i \sum_k \alpha_k \hat{s}_{k,x}} \quad \text{with} \quad \alpha_k = (1 - \epsilon_k) \frac{\pi}{2} \quad \text{and} \quad \epsilon_k = \sqrt{N} [\mathcal{H}_3^{-1}]_{jk} \quad (20)$$

and we measure, in the obtained state $\hat{U}' |\psi_t\rangle$, the optimal observable $\hat{X}$, which could be $\hat{S}_{\vec{m}}$ or $\hat{X}_{\text{MAI}}$ according to the used measurement protocol. For each $\tilde{\theta}_j$ with $j = 1, ..., N$, this procedure is repeated $\mu$ times. In the numerical simulation, the measurement results $\lambda_1, ..., \lambda_\mu$, where $\lambda_i$ is one of the eigenvalues of the measured observable, are obtained by sampling the probability distribution

$$P_i = |\langle \lambda_i | \hat{U}' | \psi_t \rangle|^2 , \quad (21)$$

where $|\lambda_i\rangle$ is the eigenstate of $\hat{X}$ associated with the eigenvalue $\lambda_i$. From these measurement results, the statistical mean $\bar{X}_\mu = \sum_i \lambda_i / \mu$ is calculated. Using the calibration curve Fig. 1(b) which gives the theoretical mean of $\langle \hat{X} \rangle$ as a function of $\Theta$ (10) or (12), the Hadamard coefficient $\tilde{\theta}_j = \sqrt{N} \Theta$ is estimated using the value of $\Theta$ for which $\langle \hat{X} \rangle = \bar{X}_\mu$. The statistical variance $(\Delta \tilde{\theta}_j)_\mu^2$ [3] is calculated numerically by repeating the procedure for estimating $\tilde{\theta}_j$ several times. Thus, all Hadamard coefficients are measured and the parameters $\theta_k$ are then deduced using (14). The scalar field (19) and its estimation with the initial state CSS, the squeezed state SSS and the state generated at $t = t_{\text{MAI,best}}$, where the measurement is performed with the MAI technique, are shown in Fig. 1(a).

## 3.2 Vector field

Let us now consider the case of a vector field discretized at $N = 8$ sites as shown in Fig. 1(c), whose unknown components $\vec{\theta}_x, \vec{\theta}_y, \vec{\theta}_z$, with $\vec{\theta}_\alpha = (\theta_{\alpha,1}, ..., \theta_{\alpha,N})^T$ for $\alpha = x, y, z$, are encoded on the atoms through the unitary evolution

$$\hat{U} = e^{-i(\hat{\vec{H}}_x \cdot \vec{\theta}_x + \hat{\vec{H}}_y \cdot \vec{\theta}_y + \hat{\vec{H}}_z \cdot \vec{\theta}_z)} , \quad (22)$$

which represents a generalization of (3) to encoding three parameters per mode. In multiparameter estimation, the measurement of parameters generated by non-commuting Hamiltonians is known to be hard, in general, because of the incompatibility of the respective optimal measurements [16, 18, 29]. This can be achieved with particular states where the expectation value of all the generator commutators is zero [18]. Here, we avoid these complications by estimating the three field components separately one after the other: first the spins are prepared in a polarized state along the $x$ direction and the measurement of the two components of the field in the $yz$-plane is performed after the OAT evolution and the application of a state rotation so as to align the optimal direction $\vec{n}$ with the $z$ or $y$ direction to measure $\vec{\theta}_z$ or $\vec{\theta}_y$, and then the spins are polarized along the $y$ direction to measure $\vec{\theta}_x$. The key point is that for the measurement of a collective linear spin observable (which excludes the estimation based on the measurement of the observable $\hat{X}_{\text{MAI}}$), the estimation of one of the field components is

---

[3]The index $\mu$ is made explicit here to remind that $(\Delta \tilde{\theta}_j)_\mu^2$ is the variance of the parameter $\tilde{\theta}_j$ deduced from $\mu$ measurements.

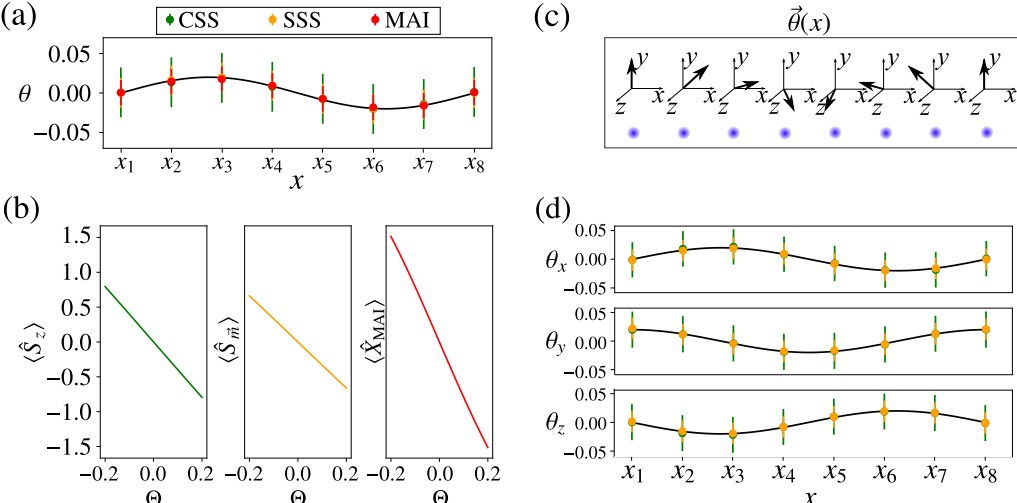

Figure 1: Numerical simulation of the estimation of a 1-dimensional field with $N = 8$ atoms: **(a)** scalar field. The field (19) with $\theta_0 = 0.02$ is represented by the solid line, and its reconstruction with $\mu = 10^3$ measurements, for each of the eight Hadamard coefficients, is represented by the symbols. The estimation is done with the spin coherent state CSS (green), the spin squeezed state SSS (orange) and the state generated at the time $t_{\text{MAI,best}}$ of the OAT dynamics where the measurement is performed with the MAI technique (red). The corresponding standard deviations (vertical lines) are obtained here by repeating 500 times the estimation procedure for each $\tilde{\theta}_j$ and they are in good agreement with the theoretical value (18). **(b)** calibration curves used with the state CSS (left), the state SSS (middle), and the state generated at $t_{\text{MAI,best}}$ (right). **(c)** and **(d)** vector field. The components (23) with $\theta_0 = 0.02$ are represented by the solid lines, and their reconstruction with the state CSS (green) and the state SSS (orange) for $\mu = 10^3$ are represented by the symbols. The vertical lines represent the corresponding standard deviations thus obtained by repeating 500 times the procedure of the estimation of each $\tilde{\theta}_j$.

not affected by the presence of the other two orthogonal components, as shown in Appendix C. In Fig. 1(d), we show the results of the estimation of the vectorial field with components

$$\theta_\alpha(x) = \theta_0 \sin(x + \varphi_\alpha), \quad \text{for} \quad \alpha = x, y, z \quad \text{and} \quad \varphi_x = 0, \; \varphi_y = \pi/2, \; \varphi_z = \pi. \quad (23)$$

# 4 Quantum gain for compressed sensing of a two-dimensional field (image)

In Sec. 2, we presented a strategy that allows us to measure a scalar signal $\vec{\theta} = (\theta_1, ..., \theta_N)^T$ through the direct estimation of the corresponding $N$ Hadamard coefficients. The estimation of each coefficient requires $\mu$ independent measurements. In this section, we will show, on a concrete example, the effect of compressed sensing, i.e. the effect of measuring only the first $L_\mathcal{H} < N$ Hadamard coefficients of a signal of size $N$, the last $N - L_\mathcal{H}$ Hadamard coefficients being taken as zero. This reduces the total number of independent measurements to be performed from $\mu N$ to $\mu L_\mathcal{H}$. Let us consider the signal 2D (the Barbara image) of size $N = 512 \times 512$ shown in Fig. 2 on the left. In the right part of the figure, the signal is reconstructed with different states of the system of $N$ atoms and for different values of $L_\mathcal{H}$. To

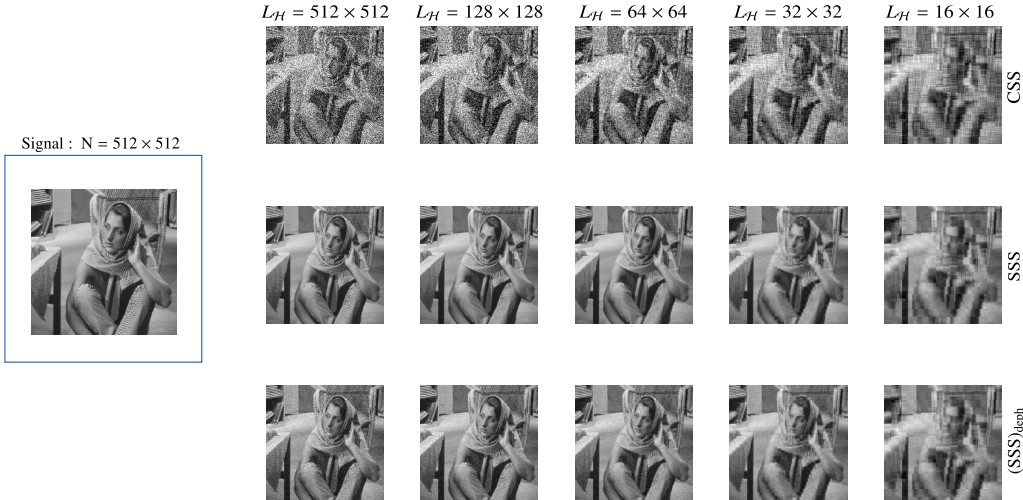

Figure 2: Example compressed sensing with quantum gain: A signal $2D$ (the Barbara image) of size $N = 512 \times 512$ (left) is reconstructed (right) with $\mu = 10$ independent measurements for each of the first $L_{\mathcal{H}} \leq N$ Hadamard coefficients, the last $N - L_{\mathcal{H}}$ coefficients being taken zero. The non-zero coeffceints are estimated with the coherent spin state CSS (top), the squeezed spin state SSS obtained by OAT dynamics (1) in the absence of decoherence (middle) and in the presence of dephasing decoherence (25) with $\gamma = 5\chi$ (bottom), for $L_{\mathcal{H}} = N = 512 \times 512$, $L_{\mathcal{H}} = 128 \times 128, 64 \times 64, 32 \times 32$, $16 \times 16$ from left to right respectively.

mimic the experimental results, we generate each of the non-zero coefficients for $j = 1, ..., L_{\mathcal{H}}$ by sampling the probability distribution

$$P(x) = \mathcal{N} e^{-\frac{(x - \tilde{\theta}_j)^2}{2(\Delta \tilde{\theta}_j)^2}}, \qquad (24)$$

where $\mathcal{N}$ is a normalization constant, $\tilde{\theta}_j$ is the $j$th Hadamard coefficient of the original image and $\Delta \tilde{\theta}_j$ is the corresponding uncertainty (18) for its estimation with a given quantum state of the $N$ spins. The first row corresponds to the CSS state (2) for which $\xi = 1$. In the second row, the state SSS is used where we have calculated the exact value of $\xi$, for the considered atom number. For the last row we have calculated the quantum gain in (18) corresponding to the $(\text{SSS})_{\text{deph}}$ state generated by the OAT evolution (1) in the presence of dephasing processes [27]

$$\frac{\partial \hat{\rho}}{\partial t} = \frac{1}{i\hbar} [\hat{H}_{\text{OAT}}, \hat{\rho}] + \gamma \left( \hat{S}_z \hat{\rho} \hat{S}_z - \frac{1}{2} \{ \hat{S}_z^2, \hat{\rho} \} \right), \qquad (25)$$

for $\gamma/\chi = 5$. Comparing the images obtained by the SSS state with those obtained with the uncorrelated CSS state, we notice that the gain due to quantum correlations is significant even with $L_{\mathcal{H}} = 32 \times 32$ (i.e. $L_{\mathcal{H}} \approx 3.9 \times 10^{-3} N$), and in the presence of decoherence. In Fig. 3, we show the results of the estimation and compression of a small signal, image of size $32 \times 32$. Also in this case, the results show a significant gain due to quantum correlations. For the results in the presence of decoherence we used the analytical formula (9) and Appendix D of Ref. [27] giving the quantum gain obtained with a OAT state in the presence of a dephasing process (25). The scaling laws of the quantum gain as a function of $N$, in the absence and presence of decoherence, are given in the same reference [27]. A comparison between the quantum gain achieved in our strategy with the maximum quantum gain given by the largest eigenvalue of the quantum Fisher information matrix is shown in Fig. 4.

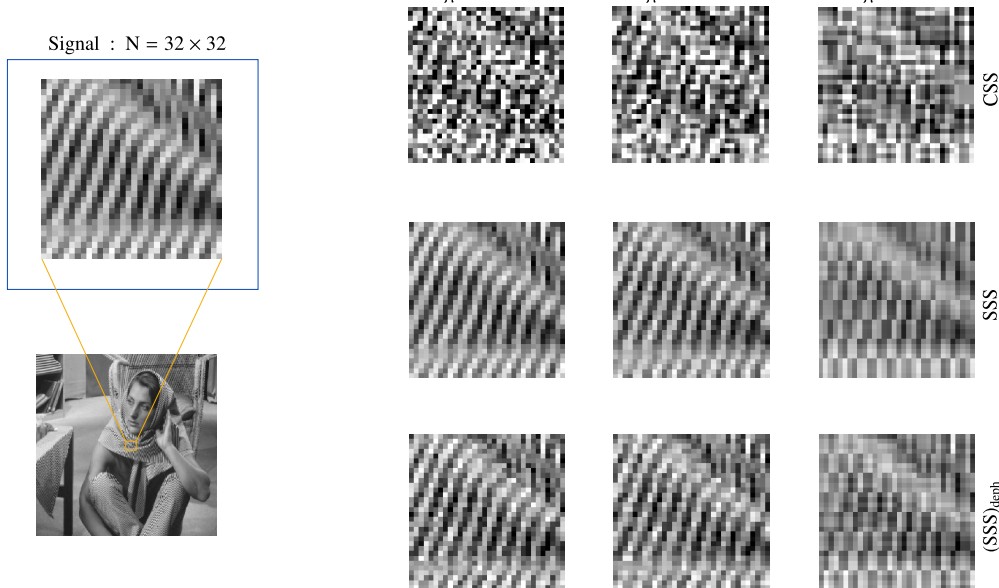

Figure 3: Example compressed sensing (small image) with quantum gain: A signal 2D (part of the Barbara image) of size $N = 32 \times 32$ (left) is reconstructed (right) with $\mu = 10$ independent measurements for each of the first $L_{\mathcal{H}} \leq N$ Hadamard coefficients, the last $N - L_{\mathcal{H}}$ coefficients being taken zero. The non-zero coefficeints are estimated with the state CSS (top), the state SSS in the absence of decoherence (middle) and in the presence of decoherence (25) with $\gamma = \chi$ (bottom), for $L_{\mathcal{H}} = N = 32 \times 32, L_{\mathcal{H}} = 16 \times 16, 8 \times 8$ from left to right respectively.

## 5 Conclusions

We have proposed a multiparameter estimation method that uses two-level atoms trapped in an optical lattice, which share internal-state quantum correlations generated by a one-axis twisting collective interaction Hamiltonian. Such a system can be obtained, for example, by adiabatically raising an optical lattice in an interacting two-component condensate (spin-squeezed Mott state) [4,5] or with fermionic atoms in a Mott-configuration in a lattice in the presence of an external laser which imprints a position-dependent phase to the atoms [6–8]. The atoms are used to measure the set of values that takes a field at the location of the different sites. The central idea of our method is that, in order to take advantage of the correlations between atoms, we measure collective quantities, the Hadamard coefficients of the signal, from which we deduce the local parameters by inverse Hadamard transformation. Although we considered the case of one atom per site, our results can be easily generalized to the case of $N$ non-interacting atoms distributed on $M$ sites with $N/M$ atoms per site. Configurations of this type can be realized by splitting a previously spin-squeezed Bose-Einstein condensate [9,10] or with cold atoms in a cavity, where cavity-mediated interactions [11] or non-local quantum non demolition measurements [12] are used to entangle the atoms in the different modes.

## Acknowledgements

We acknowledge Alan Serafin and Yvan Castin for their interface Latex-Deepl translation program for multilingual version of this paper.

### Funding information

Y.B and A.S acknowledge funding from the project macQsimal of the EU quantum flagship. M.G. acknowledges funding from Ministerio de Ciencia e Innovación (MCIN) / Agencia Estatal de Investigación (AEI) for Project No. PID2020-115761RJ-I00, support of a fellowship from "la Caixa" Foundation (ID 100010434), from the European Union's Horizon 2020 research and innovation program under Marie Skłodowska-Curie Grant Agreement No. 847648, fellowship code LCF/BQ/PI21/11830025 and Generalitat Valenciana (CDEIGENT/2021/014). This work has been financially supported by the Ministry of Economic Affairs and Digital Transformation of the Spanish Government through the QUANTUM ENIA project call - Quantum Spain project, and by the European Union through the Recovery, Transformation and Resilience Plan - NextGenerationEU within the framework of the Digital Spain 2025 Agenda. This work was partially funded by CEX2019-000910-S [MCIN/ AEI/10.13039/501100011033], Fundació Cellex, Fundació Mir-Puig, and Generalitat de Catalunya through CERCA.

## A    Change of sign of a local parameter by spin flip

In this appendix, we show that a local rotation of angle $\pi$ around the axis $x$ of a spin followed by the encoding of a parameter $\theta$ by the generator $\hat{s}_{\vec{n}}$ where $\vec{n}$ is in the $yz$-plane and another rotation of angle $-\pi$ around $x$, is equivalent to reverse the sign of the encoded parameter $\theta$. We have

$$e^{i\pi\hat{s}_x}e^{-i\theta\hat{s}_{\vec{n}}}e^{-i\pi\hat{s}_x} = \left(\sum_j \frac{(i\pi)^j}{j!}\hat{s}_x^j\right)\left(\sum_k \frac{(-i\theta)^k}{k!}\hat{s}_{\vec{n}}^k\right)\left(\sum_l \frac{(-i\pi)^l}{l!}\hat{s}_x^l\right), \qquad (A.1)$$

and

$$\begin{aligned}
\sum_k \frac{(-i\theta)^k}{k!}\hat{s}_{\vec{n}}^k &= \sum_p \frac{(-i\theta)^{2p}}{(2p)!}\hat{s}_{\vec{n}}^{2p} + \sum_p \frac{(-i\theta)^{2p+1}}{(2p+1)!}\hat{s}_{\vec{n}}^{2p+1} \\
&= \sum_p (-1)^p \frac{(\theta/2)^{2p}}{(2p)!}(2\hat{s}_{\vec{n}})^{2p} - 2i\hat{s}_{\vec{n}}\sum_p (-1)^p \frac{(\theta/2)^{2p+1}}{(2p+1)!}(2\hat{s}_{\vec{n}})^{2p} \\
&= \cos\left(\frac{\theta}{2}\right)\mathbb{1} - 2i\sin\left(\frac{\theta}{2}\right)\hat{s}_{\vec{n}}, \qquad (A.2)
\end{aligned}$$

where we used $(2\hat{s}_{\vec{n}})^2 = \hat{\sigma}_{\vec{n}}^2 = \mathbb{1}$. Replacing (A.2) in (A.1) and simplifying we find

$$\begin{aligned}
e^{i\pi\hat{s}_x}e^{-i\theta\hat{s}_{\vec{n}}}e^{-i\pi\hat{s}_x} &= \left(\cos\left(\frac{\theta}{2}\right)\mathbb{1} - 8i\sin\left(\frac{\theta}{2}\right)\hat{s}_x\hat{s}_{\vec{n}}\hat{s}_x\right) \\
&= \left(\cos\left(\frac{\theta}{2}\right)\mathbb{1} + 2i\sin\left(\frac{\theta}{2}\right)\hat{s}_{\vec{n}}\right) \\
&= e^{i\theta\hat{s}_{\vec{n}}}, \qquad (A.3)
\end{aligned}$$

where we used $8\hat{s}_x\hat{s}_{\vec{n}}\hat{s}_x = \hat{\sigma}_x\hat{\sigma}_{\vec{n}}\hat{\sigma}_x = -\hat{\sigma}_{\vec{n}} = -2\hat{s}_{\vec{n}}$, which can be demonstrated as follows:

$$\hat{\sigma}_x\hat{\sigma}_{\vec{n}}\hat{\sigma}_x = \hat{\sigma}_{\vec{n}} + \hat{\sigma}_x[\hat{\sigma}_{\vec{n}},\hat{\sigma}_x] = \hat{\sigma}_{\vec{n}} + [\hat{\sigma}_x,\hat{\sigma}_{\vec{n}}]\hat{\sigma}_x, \qquad (A.4)$$

so we deduce

$$
\begin{aligned}
\hat{\sigma}_x \hat{\sigma}_{\vec{n}} \hat{\sigma}_x &= \hat{\sigma}_{\vec{n}} + \frac{1}{2}[\hat{\sigma}_x, [\hat{\sigma}_{\vec{n}}, \hat{\sigma}_x]] \\
&= \hat{\sigma}_{\vec{n}} + \frac{1}{2}[\hat{\sigma}_x, [(n_y \hat{\sigma}_y + n_z \hat{\sigma}_z), \hat{\sigma}_x]] \\
&= \hat{\sigma}_{\vec{n}} + \frac{1}{2}[\hat{\sigma}_x, n_y[\hat{\sigma}_y, \hat{\sigma}_x] + n_z[\hat{\sigma}_z, \hat{\sigma}_x]] \\
&= \hat{\sigma}_{\vec{n}} - i\left(n_y[\hat{\sigma}_x, \hat{\sigma}_z] - n_z[\hat{\sigma}_x, \hat{\sigma}_y]\right) \\
&= \hat{\sigma}_{\vec{n}} - 2\left(n_y \hat{\sigma}_y + n_z \hat{\sigma}_z\right) \\
&= -\hat{\sigma}_{\vec{n}}.
\end{aligned}
\tag{A.5}
$$

Equation (A.3) shows that we can change the sign of an encoded parameter $\theta$ by a rotation of angle $\pi$ around the $x$ axis before encoding $\theta$ and another rotation of angle $-\pi$ around the same $x$ axis after encoding the parameter.

# B Reformulation of our protocol within the method of moments for multiparameter estimation

## B.1 Quantum Fisher information matrix and measurement optimality

In this section we study the optimality of our strategy in the absence and presence of decoherence. For this purpose, we introduce the quantum Fisher information matrix QFIM $\mathcal{F}$ which gives, in a multi-parameter estimation protocol, the limit in precision of the estimation of all parameters. In particular, $N$ parameters $\vec{\theta} = (\theta_1, ..., \theta_N)^T$, encoded on a spin state $|\psi\rangle$ via the generators $\hat{\vec{H}} = (\hat{H}_1, ...., \hat{H}_N)^T$, are estimated from the results of $\mu$ independent measurements of $N$ observables, with the covariance matrix $\Sigma$ whose elements are $\Sigma_{kl} = \text{Cov}(\theta_k, \theta_l)$. The matrix $\Sigma$ satisfies the Cramér-Rao inequality

$$
\Sigma \geq \frac{\mathcal{F}^{-1}}{\mu}.
\tag{B.1}
$$

The calculation of the quantum Fisher information matrix $\mathcal{F}$ thus allows us to evaluate the quality of the estimation strategy proposed in this paper. In the case of a pure state, such as the state prepared by the OAT dynamics in absence of noise, the QFIM is given by

$$
\mathcal{F}_{kl} = 4\text{Cov}(\hat{H}_k, \hat{H}_l).
\tag{B.2}
$$

For parameters $\vec{\theta} = (\theta_1, ..., \theta_N)^T$ encoded on a OAT state $|\psi_t\rangle$ via unitary evolution (3), with $\hat{H}_k = \hat{s}_{k,\vec{n}}$, the QFIM elements are given, to first order in all $\theta_k$ close to $\theta_k = 0$, as a function of $\vec{n}$ by

$$
\mathcal{F}_{kk} = 4(\Delta \hat{s}_{k,\vec{n}})^2 = 1 \qquad \forall\, k,
$$

$$
\mathcal{F}_{kl} = 4\text{Cov}(\hat{s}_{1,\vec{n}}, \hat{s}_{2,\vec{n}}) = \frac{4(\Delta \hat{S}_{\vec{n}})^2}{N(N-1)} - \frac{1}{N-1} \qquad \forall\, k, l \neq k.
$$

By calculating the QFIM spectrum we obtain two different eigenvalues: $\lambda_{\max}$ with multiplicity $g = 1$, where the corresponding eigenvector is $\lambda_{\max} = (1, ..., 1)^T/\sqrt{N}$, and $\lambda_{\min}$ with multiplicity $g = N-1$

$$
\lambda_{\max} = 1 + 4(N-1)\text{Cov}(\hat{s}_{1,\vec{n}}, \hat{s}_{2,\vec{n}}) = \frac{4}{N}(\Delta \hat{S}_{\vec{n}})^2,
\tag{B.3}
$$

$$
\lambda_{\min} = 1 - 4\text{Cov}(\hat{s}_{1,\vec{n}}, \hat{s}_{2,\vec{n}}) = \frac{1}{N-1}(N - \lambda_{\max}).
\tag{B.4}
$$

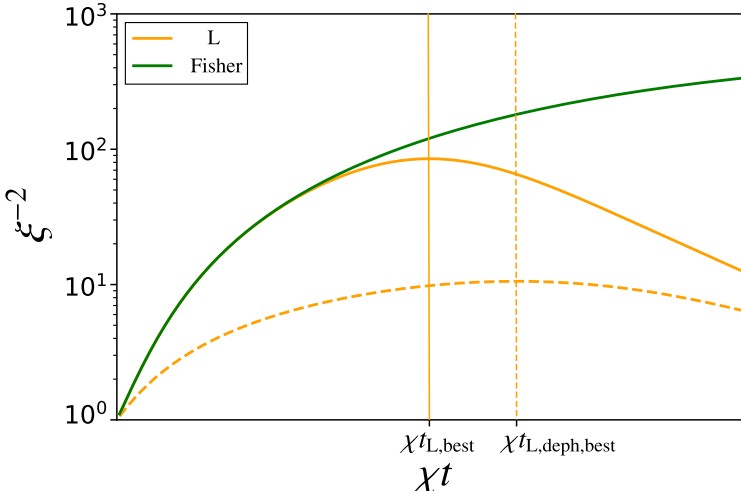

Figure 4: Metrological gain $\xi_L^{-2}$ on the estimation of each parameter combination associated with a linear measurement, in the absence (solid orange) and in the presence of decoherence described by the equation (25) with $\gamma = \chi$ (dashed orange) as a function of time compared to the optimal gain $\xi_F^{-2}$ associated with the maximum eigenvalue of the quantum Fisher information matrix (solid green) for $N = 10^3$. The vertical lines represent the optimal squeezing times.

With the direction of rotation $\vec{n}$ chosen to maximize $\lambda_{\max}$, we find $\lambda_{\max} \geq 1$ and $\lambda_{\min} \leq 1$. This shows that a state generated by the OAT dynamics can only reduce the variance of the combination of parameters $\tilde{\theta} = \vec{\lambda}_{\max} \cdot \vec{\theta}$ with $\xi_F^2/\mu \leq (\Delta \tilde{\theta})^2 \leq 1/\mu$ where $\xi_F^2 = 1/\lambda_{\max}$. As explained in Sec. 2, the estimation of other combinations is possible via the unitary evolution (9). This is equivalent to applying to $\hat{\vec{H}}$ the unitary transformation $J = \mathrm{diag}(\epsilon_1, ..., \epsilon_N)$ with $\epsilon_k = \pm 1$. The quantum Fisher information matrix becomes in this case

$$\mathcal{F}' = J \mathcal{F} J^T \,, \tag{B.5}$$

where we used $\mathrm{Cov}((J\hat{h})_k, (J\hat{h})_l) = \sum_{ij} J_{ki} J_{lj} \mathrm{Cov}(\hat{H}_i, \hat{H}_j)$. As $JJ^T = J^T J = \mathbb{1}$, the QFIM spectrum is invariant under the transformation (B.5) and the eigenvector $\vec{\lambda}'_{\max}$ of $\mathcal{F}'$ associated to the eigenvalue $\lambda_{\max}$ is given by

$$\vec{\lambda}'_{\max} = J \vec{\lambda}_{\max} \,. \tag{B.6}$$

By changing the $\epsilon_k$ in $J$ we can therefore change the optimal combination $\tilde{\theta} = (J\vec{\lambda}_{\max}) \cdot \vec{\theta}$. Thus we can independently estimate $N$ linear combinations of $\tilde{\theta}$ with a significant quantum gain. To measure the optimality of the strategy of this paper, we compare in Fig. 4. the optimal gain $\xi_F^{-2}$, associated with the Fisher information $\mathcal{F}$ optimized on $\vec{n}$, to the quantum gains associated to the OAT squeezed states (for linear measurement) in the absence and presence of decoherence (25).

## B.2 Multiparameter method of moments and optimal observable

The problem studied in our work can also be formulated within the framework of multiparameter estimation theory. Here $N$ parameters $\vec{\theta}' = (\epsilon_1 \theta_1, ..., \epsilon_N \theta_N)^T$, with $\epsilon_j = \pm 1$, are encoded by the generators $\hat{\vec{H}}_{\vec{n}} = (\hat{s}_{1,\vec{n}}, ..., \hat{s}_{N,\vec{n}})^T$ on the state $|\psi_t\rangle$ prepared by OAT dynamics for a time $t$ from CSS state (2) through the unitary evolution (3)

$$\hat{U}(\vec{\theta}) = e^{-i\hat{\vec{H}}_{\vec{n}} \cdot \vec{\theta}'} \,. \tag{B.7}$$

A change of basis of parameters $\vec{\vartheta} = P\vec{\theta}'$, with $PP^T = P^T P = \mathbb{1}$, allows to rewrite this last equation as

$$\hat{U}(\vec{\vartheta}) = e^{-i\hat{\vec{G}}_{\vec{n}} \cdot \vec{\vartheta}} \quad \text{with} \quad \hat{\vec{G}}_{\vec{n}} = P\hat{\vec{H}}_{\vec{n}}, \tag{B.8}$$

that represents the encoding of $N$ parameters $\vec{\vartheta}$ generated by observables $\hat{\vec{G}}_{\vec{n}}$. For estimating $\vec{\vartheta}$, one can use multiparameter method of moments [17] where $\vec{\vartheta}$ are estimated from the statistical means $\vec{\bar{X}}^{(\mu)} = (\bar{X}_1^{(\mu)}, ..., \bar{X}_N^{(\mu)})^T$, results of $\mu$ independent measurements of $N$ observables $\hat{\vec{X}} = (\hat{X}_1, ..., \hat{X}_N)^T$ as the values for which

$$\langle \hat{U}^\dagger(\vec{\vartheta}) \hat{X}_k \hat{U}(\vec{\vartheta}) \rangle = \bar{X}_k^{(\mu)}, \qquad k = 1, ..., N. \tag{B.9}$$

For $\mu \gg 1$, this method allows us to estimate the $\vec{\vartheta}$ with an estimator covariance matrix

$$\begin{aligned} \Sigma &= (\mu M[|\psi_t\rangle, \hat{\vec{G}}_{\vec{n}}, \hat{\vec{X}}])^{-1} \\ &= (\mu C[|\psi_t\rangle, \hat{\vec{G}}_{\vec{n}}, \hat{\vec{X}}]^T \Gamma[|\psi_t\rangle, \hat{\vec{X}}]^{-1} C[|\psi_t\rangle, \hat{\vec{G}}_{\vec{n}}, \hat{\vec{X}}])^{-1}, \end{aligned} \tag{B.10}$$

where we have introduced the commutator matrix $C[|\psi_t\rangle, \hat{\vec{G}}_{\vec{n}}, \hat{\vec{X}}]_{kl} = -i\langle [\hat{X}_k, \hat{G}_{l,\vec{n}}] \rangle$ and the covariance matrix $\Gamma[|\psi_t\rangle, \hat{\vec{X}}] = \text{Cov}(\hat{X}_k, \hat{X}_l)$. By choosing the observables $\hat{\vec{X}} = \hat{\vec{X}}_{\vec{m}} = \sqrt{N}P\hat{\vec{H}}_{\vec{m}}$, with $\vec{m}$ such that $\{\vec{m}, \vec{n}, \vec{e}_x\}$ form an orthonormal basis, the commutator matrix is given by

$$C[|\psi_t\rangle, \hat{\vec{G}}_{\vec{n}}, \hat{\vec{X}}_{\vec{m}}] = \sqrt{N}PC[|\psi_t\rangle, \hat{\vec{H}}_{\vec{n}}, \hat{\vec{H}}_{\vec{m}}]P^T = \sqrt{N}\langle \hat{s}_{1,x} \rangle \mathbb{1}, \tag{B.11}$$

where we used

$$C[|\psi_t\rangle, \hat{\vec{H}}_{\vec{n}}, \hat{\vec{H}}_{\vec{m}}]_{kl} = -i\langle [\hat{s}_{k,\vec{m}}, \hat{s}_{l,\vec{n}}] \rangle = \langle \hat{s}_{1,x} \rangle \delta_{kl} \tag{B.12}$$

and the orthogonality of $P$. Since the commutator matrix is diagonal, the system of equations (B.9) is decoupled, and the parameter $\vartheta_k$ can be estimated from the results of $\mu$ independent measurements of the observable $\hat{X}_k$ with, for $\mu \gg 1$, the uncertainty

$$(\Delta\vartheta_k)^2 = \Sigma_{kk} = \frac{1}{\mu} \frac{(\Delta\hat{X}_k)^2}{N|\langle \hat{s}_{1,x} \rangle|^2}. \tag{B.13}$$

For a given $k$ (e.g. $k = 1$), we choose $P$ so that $\hat{X}_k = \sqrt{N}\sum_l P_{kl}\hat{s}_{l,\vec{m}} = \sum_l \hat{s}_{l,\vec{m}} = \hat{S}_{\vec{m}}$, that is to say $P_{kl} = 1/\sqrt{N}$ for all $l$. With this choice of $P$ and according to equation (B.13), the combination of parameters $\vartheta_k = \sum_l \epsilon_l \theta_l / \sqrt{N}$ is estimated with the uncertainty

$$(\Delta\vartheta_k)^2 = \frac{1}{\mu} \frac{(\Delta\hat{S}_{\vec{m}})^2}{N|\langle \hat{s}_{1,x} \rangle|^2} = \frac{\xi_L^2}{\mu}. \tag{B.14}$$

This last equation is equivalent to equation (18) in the case of the measurement of a linear collective spin observable and $\vartheta_k$ is a Hadamard coefficient. Let us now consider the case of a measurement with the MAI technique, where the OAT evolution $\hat{U}_\tau = e^{-i\chi\tau\hat{S}_z^2}$ with $\tau = -t$ is applied to the system before the measurement of the $N$ observables $\hat{\vec{X}}_{\vec{m}}$, which is equivalent to measuring the observables

$$\hat{\vec{X}}_{\text{MAI}} = e^{-i\chi t\hat{S}_z^2} \hat{\vec{X}}_{\vec{m}} e^{i\chi t\hat{S}_z^2} = \sqrt{N}P(e^{-i\chi t\hat{S}_z^2}\hat{s}_{1,\vec{m}}e^{i\chi t\hat{S}_z^2}, ..., e^{-i\chi t\hat{S}_z^2}\hat{s}_{N,\vec{m}}e^{i\chi t\hat{S}_z^2})^T. \tag{B.15}$$

The commutator matrix in this case is written as

$$C\left[|\psi_t\rangle, \hat{\vec{G}}_{\vec{n}}, \hat{\vec{X}}_{\text{MAI}}\right] = \sqrt{N}PC\left[|\psi_t\rangle, \hat{\vec{H}}_{\vec{n}}, e^{-i\chi t\hat{S}_z^2}\hat{\vec{H}}_{\vec{m}}e^{i\chi t\hat{S}_z^2}\right]P^T, \tag{B.16}$$

with

$$C\left[|\psi_t\rangle,\hat{\vec{H}}_{\vec{n}},e^{-i\chi t\hat{S}_z^2}\hat{\vec{H}}_{\vec{m}}e^{i\chi t\hat{S}_z^2}\right]_{kl} = -i\left\langle\left[e^{-i\chi t\hat{S}_z^2}\hat{s}_{k,\vec{m}}e^{i\chi t\hat{S}_z^2},\hat{s}_{l,\vec{n}}\right]\right\rangle$$
$$= -i\left\langle\left[e^{-i\chi t\hat{S}_z^2}\hat{s}_{1,\vec{m}}e^{i\chi t\hat{S}_z^2},\hat{s}_{1,\vec{n}}\right]\right\rangle\delta_{kl}$$
$$- i\left\langle\left[e^{-i\chi t\hat{S}_z^2}\hat{s}_{1,\vec{m}}e^{i\chi t\hat{S}_z^2},\hat{s}_{2,\vec{n}}\right]\right\rangle(1-\delta_{kl}). \tag{B.17}$$

By looking for the matrix $P$ that diagonalizes $C[|\psi_t\rangle,\hat{\vec{G}}_{\vec{n}},\hat{\vec{X}}_{\text{MAI}}]$, we realize that for the $k$ corresponding to the maximum eigenvalue

$$C\left[|\psi_t\rangle,\hat{\vec{G}}_{\vec{n}},\hat{\vec{X}}_{\text{MAI}}\right]_{kk}^{\max} = -i\frac{\left\langle\left[e^{-i\chi t\hat{S}_z^2}\hat{S}_{\vec{m}}e^{i\chi t\hat{S}_z^2},\hat{S}_{\vec{n}}\right]\right\rangle}{\sqrt{N}}, \tag{B.18}$$

one has $P_{kl} = 1/\sqrt{N}$ for all $l$. The measurement of:

$$(\hat{X}_{\text{MAI}})_k = \sqrt{N}\sum_l P_{kl}e^{-i\chi t\hat{S}_z^2}\hat{s}_{l,\vec{m}}e^{i\chi t\hat{S}_z^2} = e^{-i\chi t\hat{S}_z^2}\hat{S}_{\vec{m}}e^{i\chi t\hat{S}_z^2},$$

thus allows to estimate the combination of the parameters $\vartheta_k = \sum_l P_{kl}\epsilon_l\theta_l = \sum_l \epsilon_l\theta_l/\sqrt{N}$ with the uncertainty

$$(\Delta\vartheta_k)^2 = \frac{1}{\mu}\frac{(\Delta\hat{X}_{\text{MAI}})^2}{|\langle[e^{-i\chi t\hat{S}_z^2}\hat{S}_{\vec{m}}e^{i\chi t\hat{S}_z^2},\hat{S}_{\vec{n}}]\rangle|^2} = \frac{1}{\mu}\frac{N/4}{|\langle[e^{-i\chi t\hat{S}_z^2}\hat{S}_{\vec{m}}e^{i\chi t\hat{S}_z^2},\hat{S}_{\vec{n}}]\rangle|^2} = \frac{\xi_{\text{MAI}}^2}{\mu}, \tag{B.19}$$

which is exactly the uncertainty (18) in the case of a MAI measurement.

## C Sequential measurement of the three components of a vector field

Here, we show how to estimate the three components $\vec{\theta}_x$, $\vec{\theta}_y$ and $\vec{\theta}_z$ of a vector field. The encoding of these components, on the state $|\psi_t\rangle$ after evolution with OAT, is done through the unitary evolution

$$\hat{U} = e^{-i\left(\vec{\theta}_x\cdot\hat{\vec{H}}_x + \vec{\theta}_y\cdot\hat{\vec{H}}_y + \vec{\theta}_z\cdot\hat{\vec{H}}_z\right)}. \tag{C.1}$$

In the vicinity of $\vec{\theta}_x = \vec{0}$, $\vec{\theta}_y = \vec{0}$ and $\vec{\theta}_z = \vec{0}$, the average of a linear collective spin observable $\hat{S}_{\vec{r}}$, with $\vec{r} = \vec{e}_y$ or $\vec{r} = \vec{e}_z$, in the state $\hat{U}|\psi_t\rangle$ is written as

$$\langle\hat{U}^\dagger\hat{S}_{\vec{r}}\hat{U}\rangle \approx -i\langle[\hat{S}_{\vec{r}},\vec{\theta}_x\cdot\hat{\vec{H}}_x + \vec{\theta}_y\cdot\hat{\vec{H}}_y + \vec{\theta}_z\cdot\hat{\vec{H}}_z]\rangle$$
$$= -i\left(\sum_{k,j}\theta_{x,k}\langle[\hat{s}_{\vec{r},j},\hat{s}_{x,k}]\rangle + \sum_{k,j}\theta_{y,k}\langle[\hat{s}_{\vec{r},j},\hat{s}_{y,k}]\rangle + \sum_{k,j}\theta_{z,k}\langle[\hat{s}_{\vec{r},j},\hat{s}_{z,k}]\rangle\right)$$
$$= -i\left(\sum_{k,j}\theta_{y,k}\langle[\hat{s}_{\vec{r},j},\hat{s}_{y,k}]\rangle + \sum_{k,j}\theta_{z,k}\langle[\hat{s}_{\vec{r},j},\hat{s}_{z,k}]\rangle\right)$$
$$= \begin{cases}\langle\hat{S}_x\rangle\left(\sum_k\theta_{y,k}/N\right), & \text{if } \vec{r} = \vec{e}_z, \\ \langle\hat{S}_x\rangle\left(\sum_k\theta_{z,k}/N\right), & \text{if } \vec{r} = \vec{e}_y\end{cases}. \tag{C.2}$$

As the average of the collective spin observable $\hat{S}_{\vec{r}}$ depends only on one component of the vector field, $\vec{\theta}_y$ or $\vec{\theta}_z$ according to the choice of $\vec{r}$, both components can be estimated separately.

By rotating the state $|\psi_t\rangle$ in order to polarize all spins along the $y$ direction using the rotation $|\psi_t'\rangle = e^{-i(\pi/2)\hat{S}_z}|\psi_t\rangle$, the average of $\hat{S}_z$ under the evolution (C.1), in the vicinity of $\vec{\theta}_x = \vec{0}$, $\vec{\theta}_y = \vec{0}$ and $\vec{\theta}_z = \vec{0}$, is given by

$$
\begin{aligned}
\langle\psi_t'|\hat{U}^\dagger\hat{S}_z\hat{U}|\psi_t'\rangle &\approx -i\sum_{k,j}\theta_{x,k}\langle\psi_t'|[\hat{s}_{z,j},\hat{s}_{x,k}]|\psi_t'\rangle \\
&= \langle\psi_t'|\hat{S}_y|\psi_t'\rangle\left(\sum_k\theta_{x,k}/N\right) \\
&= \langle\hat{S}_x\rangle\left(\sum_k\theta_{x,k}/N\right).
\end{aligned}
\tag{C.3}
$$

The measurement of $\hat{S}_z$ in this case allows us to estimate the $\vec{\theta}_x$ component of the field. Thus, we measure a vector field.

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
