# Peer review of "Quantum-enhanced multiparameter estimation and compressed sensing of a field"

_SciPost Physics, doi:SciPost Phys. 14, 050 (2023)_

## Round 1 · Referee Report · Jing Liu (Referee 1) · 2022-8-28

Report

Quantum multiparameter estimation is a core problem in quantum parameter estimation, and the performance of a quantum system on multiparameter estimation could dramatically different with the single-parameter counterpart. Hence, the study of multiparameter estimation in various quantum systems or scenarios and design of schemes that present quantum advantages are very essential. The authors study a multiparameter estimation problem in a collective spin system, and provide a scheme for the measurement of the linear combinations of a set of parameters. The paper is interesting and would benefit the scientists working on both quantum parameter estimation and collective spin systems. I think I can recommend it to be published on SciPost. Hereby are my suggestions and comments.

  1. The entire analysis in this paper is based on the measurement of an observable, and the corresponding transfer function is up to first order. In quantum parameter estimation, a useful tool to analyze the theoretical performance of the multiparameter estimation is the quantum Cramer-Rao bound [more details could be found in a recent review: J. Phys. A: Math. Theor. 53 023001 (2020)]. Of course this tool has some drawbacks like it could be unattainable, but the comparison between it and Eqs. (6) and (13) would give the readers some impressions how good the provided scheme is. I suggest the authors calculate the quantum Fisher information matrix in this case and do some comparison.

  2. In the provided scheme, the parameters are not measured directly, but calculated via linear combinations. As a matter of fact, the performance of different types of linear combinations is different. From the perspective of quantum Cramer-Rao bound, the quantum Fisher information matrix of linear combinations and the original parameters are connected via the Jacobian matrix. Hence, I am curious that whether there exists a set of $epsilon_k$ that gives the minimum total variance of the original parameters. This analysis may also tell the readers if the measurement of Hadamard coefficients is optimal in theory.

  3. The discussion of quantum imaging in Sec. 4 is very interesting. Yet, there are not many technical details like how to calculate the dynamics numerically or analytically in Eq. (25) in the case of L_H=512*512. The total Hilbert space seems very large in this case. I think the presentation of more technical details would reduce the difficulty of repeating the results in the paper, and help the readers to better understand the corresponding meaning.

  • validity: high
  • significance: good
  • originality: high
  • clarity: high
  • formatting: excellent
  • grammar: excellent

Author:  Youcef Baamara  on 2022-09-28  [id 2862]

(in reply to Report 1 by Jing Liu on 2022-08-28)

Answer to question 1:
Indeed, the quantum Fisher information matrix is important for evaluating the quality of the metrological gain obtained in a multiparameter estimation scheme. We added subsection B.1 in Appendix B where we calculate the quantum Fisher information matrix. Its maximum eigenvalue, which gives the optimal quantum gain that could be achieved by the OAT states, is then compared in figure 4 with the metrological gain obtained by our strategy in the absence and presence of decoherence.

Answer to question 2:
In subsection B.1 of appendix B we obtained that for a state generated by the OAT dynamics, the quantum Fisher information matrix admits a single combination of parameters that achieves a quantum advantage. Thus, for a given state, only one Hadamard coefficient can be obtained with a metrological gain. In order to obtain another coefficient we need to change the state via the local flips of the individual spins as shown in the text. A change of the $\epsilon_k$ to select a given Hadamard coefficient is achieved via a transformation that leaves the spectrum of the quantum Fisher information matrix invariant.

Answer to question 3:
The details of the calculation of the squeezing dynamics described by equation (25) are presented in a previous paper. We have added a sentence at the end of section 4 where we make a precise reference to the formulas that we use from that reference.

---

## Round 1 · Referee Report · Anonymous (Referee 2) · 2022-9-4

Report

see attachement

Attachment

  • validity: high
  • significance: good
  • originality: good
  • clarity: high
  • formatting: excellent
  • grammar: excellent

Author:  Youcef Baamara  on 2022-09-28  [id 2861]

(in reply to Report 2 on 2022-09-04)

Answer to questions 1 and 2:
In multiparameter estimation theory, it is in general the matter of estimating N parameters by means of repeated measurements of $N$ observables of the system starting from the same quantum state (the observables can possibly be measured simultaneously in one realization of the experiment in case they commute). In this framework, the covariance matrix of the estimators and the quantum Fisher matrix $\cal F$, which are related by the multiparameter Cramér-Rao inequality, are usually introduced. However, as we now show it in the appendix B.1, for a one-axis-twisting squeezed state it is not possible to obtain a quantum gain in the estimation of each parameter by this strategy. For a given quantum state, only one eigenvalue of $\cal F$, corresponding to a particular combination of the parameters shows a quantum advantage when all the others show a disadvantage. To obtain a quantum advantage in another combination it is necessary to change the state. To estimate $N$ combinations with a quantum advantage, $N^2$ measurements should then be performed. In our strategy on the contrary, for a given state only one collective observable must be measured to estimate the combination that shows a quantum advantage. To estimate the $N$ combinations, it is sufficient to measure $N$ collective observables.

Answer to question 3:
Analytical results concerning the metrological gain and its scaling with $N$ are discussed in detail in an earlier article. To point this out, we have added a sentence at the end of section 2 that shows the reader where to find the technical details.

Answer to question 4:
A sentence was added in the introduction to cite previous works in multiparameter estimation and a footnote to explain the considered frame and results of the cited works and to show the advantage of our strategy.

---

## Round 2 · Referee Report · Anonymous (Referee 2) · 2022-10-1

Report

I am a bit puzzled by the authors' response on the advantage compared to the local measurement. It seems we may not be talking about the same local strategy. Take the case of estimating the scaler field at N sites, $\theta=(\theta_1, ...,\theta_N)$ ($\theta_j$ is at site j) for example, the local strategy is to prepare a squeezed state with N spins at one site(for example, N spins at site j) and estimate one parameter(for example $\theta_j$) at one time, then repeating the strategy at different sites to get the estimation of all N parameters. For the estimation of $\theta_j$ at site j, this is just a single-parameter problem and we have only quantum Fisher information, not quantum Fisher information matrix. So it is not clear how the eigenvalues of quantum Fisher information matrix come into play in the local strategy for the case of scaler field. Considering the sensitivity, for the local strategy with N spins prepared as the squeezed state at site j, the precision scales as $1/N^2$ for the estimation of $\theta_j$, and this has to be repeated N times to get all N parameters. For the collective strategy with N spins the precision also scales as $1/N^2$ for the estimation of one Hardmard coefficient and this also has to be repated N times to get all N Hardmard coeffi cients. There does not seem to be a difference on the precision.

I will appreciate if the authors can make a further clarification on the advantage.
  • validity: high
  • significance: good
  • originality: good
  • clarity: good
  • formatting: good
  • grammar: excellent

Author:  Youcef Baamara  on 2022-10-12  [id 2918]

(in reply to Report 1 on 2022-10-01)

1-Framework 1.1- Our Strategy: In our work, we consider a configuration in which $N$ atoms distributed in spatially separated modes, for example at the nodes of an optical lattice (1 atom per site), are used to estimate $N$ parameters that are for example the values taken by a spatially extended field. As a result, the use of a coherent spin state (CSS), with $N$ collective measurements, allows us to estimate all the parameters with a variance (for one repetition of the measurement $\mu=1$) \begin{align} (\Delta\theta_k)_{\rm CSS}=1\qquad\forall k. \end{align} Quantum correlations, generated by the one axis twisting (OAT) dynamics, with $N$ collective measurements, allow for a quantum enhancement where all the parameters are estimated with variance (for one repetition of the measurement $\mu=1$) \begin{align} (\Delta\theta_k)=\xi(N,t)\qquad\forall k. \end{align} $\xi(N,t)$ is the squeezing parameter associated to the state generated by OAT at time $t$ for single parameter estimation.

1.2- The " Scanning Microscope " strategy: The "local strategy" mentioned in your report, is the one that we called in the introduction "Scanning Microscope", where one would use a set of $N$ atoms to locally estimate the field at each site. In this strategy, a coherent spin state, with collective measurements, allows us to estimate all the parameters with a variance (for one repetition of the measurement $\mu=1$) \begin{align} (\Delta\theta_k)_{\rm CSS}=\frac{1}{\sqrt N}\qquad\forall k. \end{align} Quantum correlations, generated by the one axis twisting (OAT) dynamics, with $N$ collective measurements, allow for a quantum enhancement where one obtains \begin{align} (\Delta\theta_k)=\frac{\xi(N,t)}{\sqrt N}\qquad\forall k. \end{align}

2- Comparison of the two strategies

  • The quantum gain due to correlations generated by OAT dynamics, $\xi(N,t)$, is the same in both strategies.
  • The variance of the estimated parameters $\Delta\theta_k$ is not the same where the "Scanning Microscope" strategy has an advantage of a statistical factor $1/\sqrt N$. This comes from the fact that each parameter is encoded on an ensemble of $N$ atoms in the "Scanning Microscope" strategy while, in our strategy, each parameter is encoded on only a single atom.
  • In the strategy we consider all the atoms remain in fixed positions and do not interact while the "Scanning Microscope" strategy requires physically moving all the atoms constituting the sensor, which exposes it to problems related to the interaction between atoms: "over squeezing" and atom loss, and, if one wants to keep the ensemble diluted, it reduces the spatial resolution of the sensor.
  • Another advantage of our strategy is that it naturally allows for "compressed sensing" by measuring only the first $L_{\cal H}<N$ Hadamard coefficients of the discretized field on the lattice, as we show in figure 5. This, in contrast, is not allowed in the "Scanning Microscope" strategy.

3- Further modifications of the paper As a further change, we have expanded the paragraph in the introduction where we compare to the "Scanning microscope" approach to fully clarify this point.

---

## Round 2 · Referee Report · Jing Liu (Referee 1) · 2022-10-7

Report

I am convinced by the authors' reply and revised manuscript and I recommend it to be published on SciPost now.

---

## Round 2 · Author Response

We thank the referees for the useful comments. To answer to them, we modified the paper by adding an appendix (Appendix B.1), a figure (Figure 4), one footnote and several sentences throughout the text, which we think improves the paper. We also included a few useful new references.

---

## Round 2 · List of Changes

• New affiliation for one of the authors (M.G.)
  • A sentence was added in the introduction to cite previous works in multiparameter estimation (to which we added the references [12], [13] and [19]) and a footnote to explain the considered frame and results of the cited works and to show the advantage of our strategy.
  • In the end of section 2, we added a sentence explaining where to find the technical details on the metrological gain and its scaling with N that were discussed in an earlier article.
  • In subsection 3.2 we make a reference to the work of Goldberg [17].
  • In the end of section 4, we added a sentence where we make a precise reference to the formulas needed to the calculation of the squeezing dynamics in the presence of decoherence described by equation (25).
  • M.G. added funding informations to acknowledgments.
  • We added subsection B.1 in Appendix B where we calculate the quantum Fisher information matrix. Its maximum eigenvalue, which gives the optimal quantum gain that could be achieved by the OAT states, is then compared in figure 4 with the metrological gain obtained by our strategy in the absence and presence of decoherence.

---

## Round 3 · Referee Report · Anonymous (Referee 2) · 2022-10-28

Report
I appreciate the authors' detailed response. The authors has now mentioned that the strategy can not achieve high precision than the local strategy. But I am also not sure the strategy gives up the $1/\sqrt{N}$ scaling as the authors now state. It seems during the comparison the authors consider the local operator $s_j,x$ and the global $S_x$ in the same way. They are, however, different. The global operator has larger gap between the maximal and minimal energy level, thus higher precisions can be achieved for the estimation of its coefficient.
Another advantage the authors claimed is that "ours has the advantage that a single collective measurement has to be performed...all
measurements in our protocol are collective". It is not quite clear what the advantage is here. Collective measurement is in general harder to perform, 'all measurements are collective' does not sounds like an advantage. Do the authors mean "all measurements are the same"?
Another advantage the authors claimed is that "ours has the advantage that a single collective measurement has to be performed...all
measurements in our protocol are collective". It is not quite clear what the advantage is here. Collective measurement is in general harder to perform, 'all measurements are collective' does not sounds like an advantage. Do the authors mean "all measurements are the same"?

Author: Youcef Baamara on 2022-11-03 [id 2980]
(in reply to Report 1 on 2022-10-28)1- Indeed, the referee is right « the collective observable has larger gap between the maximal and minimal eigenvalue », and this is already taken into account in our calculation of sensitivity (see appendix B.2). In fact, in the multiparameter estimation schemes, $N$ different phases $\theta_k$ are encoded in the system via local generators (for us the individual generators $\hat s_{\vec{n},k}$). It is only in the particular case where all the phases are equal ($\theta_k=\theta,\:\forall k$) that we find the statistical factor $1/\sqrt{N}$ corresponding to the case of estimation of a single parameter $\theta$ encoded by the collective generator $\hat{S}_{\vec{n}}$.
2- Let us now come back to the case where the atoms are distributed in spatially separated modes. Unlike other multiparameter estimation schemes that require local measurements (i.e. measurements on individual atoms), our scheme requires only collective measurements. By "collective measurements" we mean measurements of the components of the collective spin operator (e.g. $\hat S_x$, $\hat S_y$ or $\hat S_z$), which indeed implies that the same spin component is measured for each atom. These collective measurements are naturally and usually performed in cold atoms experiments. What makes these measurements advantageous is the fact that it is not necessary to have the spatial resolution that would be required to perform measurements on individual atoms (i.e. local measurements).
Changes to the manuscript: we have added a footnote (footnote 1) to explain what we mean by a collective measurement, and a sentence in the end of the fourth paragraph of the introduction to explain its advantage with respect to local measurements.

---

## Round 4 · Referee Report · Anonymous (Referee 2) · 2022-11-4

Report

I recommend it to be published now.

---

## Round 4 · List of Changes

• We have added a footnote (footnote 1) to explain what we mean by a collective measurement,

  • We have added a sentence in the end of the fourth paragraph of the introduction to explain the advantage of collective measurements with respect to local measurements.

---

## Editorial Decision

published